# PATHS: Parameter-wise Adaptive Two-Stage Training Harnessing Scene Transition Mask Adapters for Video Retrieval

## Abstract

Image-text pre-trained model, e.g., CLIP, has gained significant traction even in the field of video-text learning. Recent approaches extended CLIP to video tasks, and have achieved unprecedented performances in the foundational study of video understanding: text-video retrieval. However, unlike conventional transfer learning within the same domain, transfer learning across different modalities from images to videos often requires fine-tuning the whole pre-trained weights rather than keeping them frozen. This may result in overfitting and distorting the pre-trained weights, leading to a degradation in performance. To address this challenge, we introduce a learning strategy, termed Parameter-wise Adaptive Two-stage training Harnessing Scene transition mask adapter (PATHS). Our two-stage learning process alleviates the deviations of the pre-trained weights. A novel method of finding the optimal weights is used in the first stage, which efficiently narrows down to strong candidates by only monitoring the fluctuations of parameters. Once the parameters are fixed to optimal values, the second stage is dedicated to acquiring knowledge of scenes with an adapter module. PATHS can be applied to any existing models in a plug-and-play manner, and always achieves performance improvements from the base models. We report state-of-the-art performances across key text-video benchmark datasets, including MSRVTT and LSMDC. Our code is available at `https://anonymous.4open.science/r/PATHS_`.

## 1 Introduction

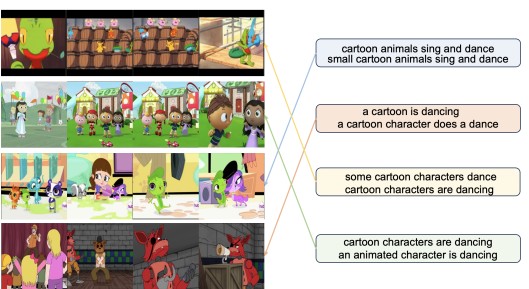

Figure 1: The video and text data examples are from the MSRVTT dataset. True pairs of video-text are denoted with links. This example illustrates the potential confusion in the learning process due to the gap between visual and textual representation.

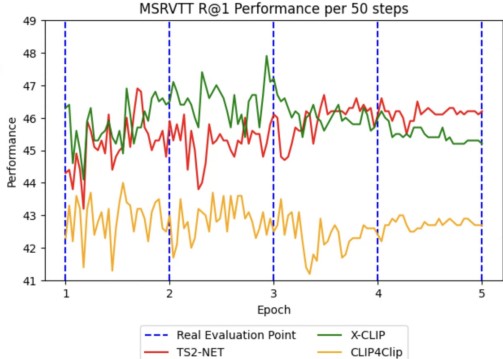

Figure 2: Recall at 1 is computed at every 50 steps in validation set from MSRVTT dataset. It displays the results of evaluating existing models every 50 steps while training.

The pre-trained network trained on image-text pairs, known as CLIP (Radford et al., 2021), has shown a significant impact on AI research. Through fine-tuning, CLIP has been extended from image-text multimodal tasks to broader tasks such as video retrieval (Luo et al., 2022; Zhao et al.,

2022; Gorti et al., 2022; Liu et al., 2022; Ma et al., 2022). When performing transfer learning within the same domain, many methods try to preserve the performance of the pre-trained model by performing linear probing. However, due to the difference between image and video, the pre-trained model image domain cannot be directly adopted into tasks involving video. The predominant method here is to take informative frames as images, and to relate them with a well-trained model: CLIP. Each frame in a video is essentially an image without *temporality*, and thus multimodal models for video understanding try to learn additional "temporal information" from the sequence of frames. Due to the huge gap between video and image, it involves fine-tuning the entire pre-trained weights by funneling temporal information to the existing model knowledge.

Many existing models in video understanding adopt this approach, adding each model's methods to capture temporal information. These models use a method of embedding information about each modality in a joint latent space and aligning the two domains. However, due to the information asymmetry among image, video, and text, fine-tuning can harm the well-trained weights from CLIP, which causes performance degradation. For instance, the ambiguity raised from nonspecific or generic text may cause confusion during learning. Looking at Figure 1, all the clips on the left seem to be relevant to all of the text captions on the right. In fact, each clip on the left is only matched to one of the text on the right, where the true pairs are connected with links. This becomes problematic as the positive text and negative text are too similar, which causes the model to be confused during training. Another observation on each true pair is that the corresponding text caption only matches with a specific scene with small sub-frames. This also brings performance degradation when irrelevant scenes are also unexpectedly aligned with the corresponding text under the matched pair. Both phenomena, mainly from the nature of the video, negatively impact the text-video models. This phenomena is also reflected in Figure 2, where existing models exhibit unstable performance in validation set during training. At each end of epoch, the performance tend to be worse than other steps within the same epoch, and optimal performance cannot be reached.

Several recent studies (Gorti et al., 2022; Jin et al., 2022; 2023b;a) have begun to use frequent evaluations at every $N$ step to improve generalization, different from the conventional neural training method, which performs one evaluation at each epoch. While this approach can potentially improve performance, it incurs extra computational overhead, indicating a need for a more efficient solution.

In light of the issues above, this study introduces the Parameter-wise Adaptive Two-stage training Harnessing Scene Transition with Masked Adapters (PATHS). Instead of directly fine-tuning the model, which often distorts the pre-trained weights, our approach adopts a two-stage method that incorporates the essence of transfer learning. The first stage of PATHS focuses on optimizing the pre-trained weight for the image-to-video modality. It identifies the points where the network parameters fluctuate and evaluates those points, enhancing the model's latent ability without needing $N$-step evaluations. PATHS is designed to achieve optimal performance efficiently based on the given metric by indirectly monitoring how the model performs based on parameter group changes. This ensures that the pre-trained weights from the image-text modality are minimally compromised when extended to the video-text modality. Subsequently, in the second stage, we introduce an adapter module that enables the model to acquire knowledge about different scenes, a pivotal aspect of video understanding. This module not only enhances the model's ability to focus on the relevant scene in a video but also fosters a deeper understanding of the scene transitions. By adopting this two-stage approach, our model strives to learn a video from diverse perspectives, maximizing its potential and offering a robust solution to the challenges identified in the existing literature. Our contributions can be summarized in three ways:

- We propose a parameter-wise two-stage training strategy to efficiently mitigate the weight corruption problem that arises from extending pre-trained weight of CLIP to tasks in the video domain. The proposed training scheme is general and thus applicable to any existing transfer learning models.

- An adapter module is additionally proposed and incorporated into our proposed scheme to distinguish different scenes, which let the model focus on relevant scene not the whole video frames during training.

- Our proposed scheme has been applied to strong baselines in a plug-and-play manner, where we show constant performance improvements achieving state-of-the-art performance in the text-to-video retrieval task with two popular datasets: LSMDC (Rohrbach et al., 2015) and MSRVTT (Xu et al., 2016)

## 2 RELATED WORKS

### 2.1 TEXT-TO-VIDEO RETRIEVAL

Text-to-video retrieval has emerged as a crucial avenue for exploring video-language comprehension. Researchers have presented an array of methods for probing the interconnection between video and language. Early studies (Gabeur et al., 2020; Croitoru et al., 2021; Li et al., 2020; Sun et al., 2019; Xu et al., 2021; Patrick et al., 2020) gravitated towards leveraging a combination of pre-extracted features from specialized models (Xie et al., 2017; Feichtenhofer et al., 2019; Devlin et al., 2018; Liu et al., 2019). This line of work has been followed by ClipBERT (Lei et al., 2021) and Frozen (Bain et al., 2021), and the idea of uniform sampling of a clip from Frozen has been widely adopted in many subsequent models. With the introduction of the large-scale image-text pre-trained model CLIP (Radford et al., 2021), CLIP4Clip (Luo et al., 2022) introduced how CLIP can be transferred to tasks for video, and has achieved unprecedented performances in tasks in video representation learning. Many following studies (Zhao et al., 2022; Gorti et al., 2022; Bogolin et al., 2022; Liu et al., 2022; Ma et al., 2022; Kang & Cho, 2023; Jin et al., 2023a;c;b; Wu et al., 2023) have developed a range of techniques grounded in the framework of CLIP4Clip. In addition to the method of transfer learning the pre-trained weight of CLIP, a foundation model for pre-training a large amount of video is also actively being studied (Wang et al., 2023; Xue et al., 2022; Alayrac et al., 2022; Yuan et al., 2021; Huang et al., 2023).

Previous research within the domain of video-language understanding has concentrated on applying pre-trained weights of CLIP to video modality and investigating the learning for video temporality. However, most existing studies suffer from the overfitting problem which is caused by the deformation of pre-trained weights of CLIP. This shortcoming can further compound the complexity of the learning process, tailoring it excessively to the given data and potentially undermining the model's generalizability. Consequently, we may benefit from examining these challenges, fostering the development methods and mechanisms that balance the intricate interplay between pre-trained model utilization and specificity.

### 2.2 ADAPTER METHOD

Adapters are specialized components or modules integrated into neural networks, particularly in transformer architectures, to facilitate task-specific adaptations without requiring extensive retraining. By introducing these specialized modules within transformer encoders, adapters enable fine-tuning for particular tasks, aiding pre-trained models to adapt to downstream NLP tasks (Stickland & Murray, 2019; Houlsby et al., 2019). They are also utilized as a cost-saving strategy in scenarios where assembling large-scale datasets is challenging. In the field of computer vision, adapters have been used for progressive learning (Rebuffi et al., 2017; Chen et al., 2022), and methods like CLIP (Radford et al., 2021) have been employed to foster zero-shot transferable features across diverse image categorization tasks. Notably, following the emergence of CLIP, a multitude of CLIP-based adapters have been launched (Sung et al., 2022; Zhang et al., 2021; Li et al., 2021), enhancing the practicality of pre-trained knowledge for few-shot downstream operations.

In the field of video retrieval, a plethora of recent studies have actively engaged in various adapter research. Some investigations (Zhang et al., 2023a; Lu et al., 2023) conducting pre-training on video text have achieved remarkable performance by incorporating adapter modules into text or video encoders without substantially increasing parameters or retraining. Similarly, several studies (Zhang et al., 2023a;b; Cheng et al., 2023) transferring the pre-trained weights of CLIP have yielded favorable outcomes by adding adapter modules. However, most of the adapters proposed in previous studies are model dependent, where as the adaptor we propose is model-agnostic.

## 3 PRELIMINARIES

Before explaining the proposed method, we briefly describe the backbone structure which extends the pre-trained weight of CLIP to the video-text retrieval task. Generally, many of the existing methods for text-to-video retrieval employing pre-trained weights of CLIP use CLIP4Clip as the backbone model. We therefore briefly explain the method of CLIP4Clip:

The video retrieval task involves embedding each modality into a joint latent space for video-text alignment. Specifically, the similarity score between a given pair of video-text is used for the alignment. The model performs feature extraction from videos and texts using pre-trained weights of CLIP. The methodology is configured as follows: Vision Transformers (Dosovitskiy et al., 2021) (ViT) are utilized to extract visual features from frames and provide them as frames for the input video. There are 12 layers, using pre-trained weights from CLIP. Text features are handled similarly to CLIP, using a transformer consisting of 12 layers. Both transformer layer contains eight heads.

### 3.1 MODALITY ENCODERS

We use encoders from CLIP to extract features for each modality. The text encoder receives the text representation $T$ as input. Here $T = \{w_{cls}; w_1, w_2, w_3, ..., w_L\}$, where $w_j$ is the $j$-th word token, and $L$ is the number of word tokens. Visual encoder is used for extracting each frame embedding, where the video representation $V$ is fed as input. The visual encoder takes the form of $v_i = \{p_{cls}; p_i^1, p_i^2, p_i^3, ..., p_i^P\}$ from ViT, where $P$ is the number of patches and $p$ is patches from the ViT encoder, and $v_i$ is the patch level embedding of each video. Finally, we represent the input video as $V = \{v_1, v_2, v_3, ..., v_m\}$ with $m$ being the maximum number of frames in a video.

### 3.2 TRAINING OBJECTIVE

Utilizing the pre-trained weights from CLIP, the model is optimized through fine-tuning these weights. We perform optimization using the similarity score $S(V, T)$, where the given video-text pair is compared with cosine similarity;

$$S(V, T) = \frac{V}{||V||}^{\top} \frac{T}{||T||}. \tag{1}$$

The optimization of this equation is performed by minimizing the symmetric cross-entropy loss shown below:

$$\mathcal{L}_V = -\frac{1}{B} \sum_{i=1}^{B} \log \frac{\exp(S(V_i, T_i))}{\sum_{j=1}^{B} \exp(S(V_j, T_i))}; \qquad \mathcal{L}_T = -\frac{1}{B} \sum_{i=1}^{B} \log \frac{\exp(S(V_i, T_i))}{\sum_{j=1}^{B} \exp(S(V_i, T_j))}, \tag{2}$$

where $B$ is the batch size. The equation below is the final objective function.

$$\mathcal{L} = \mathcal{L}_V + \mathcal{L}_T. \tag{3}$$

## 4 METHODOLOGY

We present our two-stage training strategy specifically designed to extend text-image pre-trained weights into the text-video modality. This approach is different from the conventional method with direct fine-tuning which often causes overfitting or strong deviations from the pre-trained weights. Our approach aims to achieve stable and effective knowledge transfer. We start with elaborating the two-stage training scheme in Section 4.1. In Section 4.2, we introduce the Scene Transition Mask Adapter (STMA) module which enables our model to learn different information from different scenes to better align text to corresponding scenes.

### 4.1 PARAMETER-WISE TWO-STAGE TRAINING METHODS

Existing studies in text-video retrieval have tested the validation set at every end of the epoch and compared their performances following the deep learning convention. Some recent models started to evaluate models more frequently within an epoch, such as at every $N$-step, and achieved further performance improvements. This remedy turns out to be effective (see top two rows in Table 4) when performing transfer learning in multimodal data, e.g., text and video. However, while effective, performing evaluation at every $N$-step leads to computational overhead (see Appendix A.1). Besides, $N$ becomes a hyperparameter, which is not given beforehand. In studies with this remedy, often the hyperparameter $N$ is not reported, nor using this evaluation technique is not revealed in the papers. Building on these observations and the need for a more efficient approach, we introduce a novel Parameter-wise Two-stage Training Method to find an optimal initialization point for our two-stage strategy. The proposed method applies to any existing models in the literature.

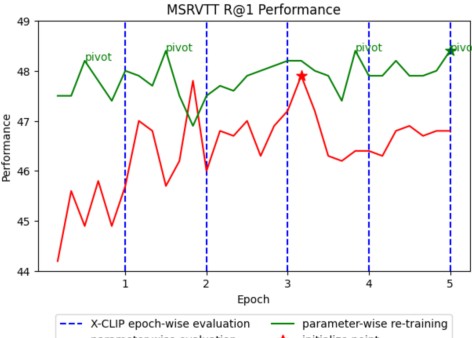

Figure 3: In the second stage, we use the best performing points in each epoch as pivots.

| Method | Text-to-Video Retrieval | | | |
|---|---|---|---|---|
| | R@1 | R@5 | R@10 | MeanR |
| One-stage scheme | | | | |
| Baseline | 47.2 | 73.5 | 82.5 | 13.8 |
| + STMA | 47.4 | 73.2 | **82.7** | **13.2** |
| Two-stage scheme | | | | |
| PATHS | **48.4** | **73.7** | 82.7 | 13.2 |
| w/o STMA | 48.1 | 72.7 | 82.5 | 13.9 |
| w/o Freeze | 47.8 | 73.3 | 82.3 | 13.7 |
| w/o Pivot | 48.2 | 73.6 | 82.4 | **13.2** |

Table 1: Results on MSRVTT dataset with varying the components in PATHS. We use X-CLIP as a baseline.

**Motivation.** Our main idea is motivated by the results in Figure 2, where the performance at every end of the epoch is often worse than most other steps within a given epoch. Here, the validation set is evaluated at every $50$ steps. The end of each epoch is denoted in dotted vertical lines. It can be clearly seen that the performance of the validation set is noisy and the achieved performance at the end of each epoch is far from optimal. As such, one obvious solution for improving the performance than comparing the performance at every end of the epoch is to compare the performance at every step. However, this requires exhaustive computations to compare validation performances at every step even with minimum changes. One might perform a *compromised* evaluation at every $N$-step for efficiency, which is an ad-hoc solution. While larger $N$ brings computational efficiency, the optimal parameters with the highest performance can be missed with higher chances. This dilemma has stimulated investigations into alternative solutions. The solution we proposed is based on our key observation. When the model converges, parameters in the neural network exhibit stable behavior. Here, we focus on the other way, i.e., when the model diverges. When the model starts to diverge after passing the optimum point, the model parameter values exhibit strong fluctuations. This often involves rearranging parameters in terms of importance (or the value of parameters). We conversely use this ranking dynamics to capture strong candidates which can bring best performance. Namely, when there's a strong change in the ranking of parameter groups at step $n$, we keep the parameter set from step $(n-1)$ as a candidate. Thus, we can take top-$K$ candidates at each epoch for further comparison.

**Main Idea.** Throughout the study, we fix $K = 5$ for our parameter-wise method, where we keep top-5 candidates which are associated with top-5 highest dynamics. These top-5 are compared in the following steps. The red graph in Figure 3 compares the performance in R@1. In accordance with Figure 2, we denote end of each epoch with dotted line. It is clearly visible that the parameter-wise method we propose in this study can effectively capture strong candidates with higher performance compared against at each end of epoch. This observation also confirms that the model achieves better performance after the model parameters are stabilized. Among all the candidates, we find the best performance (denoted as star in Figure 3), where the set of parameters are saved for later use in the second stage.

**Quantifying Dynamics** The parameter-wise two-stage training method captures the moment when the parameter is not stabilized, stores the parameters in memory, and evaluates them at the end of each epoch. However, parameters behave unpredictably, making it challenging to compare how parameters behave unstable. To address this, we group the parameters and evaluate the stabilization level based on each parameter group's ranking dynamic. For measuring the ranking dynamics, we group the parameters according to their functionalities (e.g., visual, textual, temporal, and etc.), and rank the parameters at each group with respect to their average fluctuations. Each ranking is stored and compared with the ranking of the average change in the next step. By doing this, we can monitor how the parameter groups have changed compared to the previous step. By continuously monitoring the rank-based fluctuations in parameter changes and employing statistical metrics such as moving averages and standard deviations, we are able to discern the underlying trends governing parameter group behavior. Utilizing this analytical framework, we evaluate the stability of the cur-

rent parameter configurations by contrasting the observed ranking shifts at the current iteration with these foundational trends. In this study, we further investigated three of our approaches, which is elaborated and compared in Section 5.4

**Two-stage Process.** In addition to finding the optimal step, we further apply the second process to maximize the model performance. Thus, the final form of our proposed parameter-wise method is developed upon the two-stage process, where the second stage involves our pivot method. The green graph in Figure 3 shows the results of our second stage of training. At the end of first stage, we initialize the model parameters with values obtained from the best performing step marked with red stars, and freeze all the parameters for second stage. In the second stage, we add an adapter (see Section 4.2) for further training. We keep the parameters at the point where the validation performance is the highest. We then load these parameters back into the model at the end of each epoch to perform the pivoting. This method prevents overfittings while maximizing the performance of the model.

Readers may have some questions on this method. The first question may arise as to whether our approach improves performance simply by adding parameters, not from the two-stage training process. This concern seems logical, but it can be refuted through the empirical evidence in Table 1. Our study found that without using our parameter-wise two-stage training method, but merely adding an adapter module and trained end-to-end, the performance was worse than our model (PATHS). This demonstrates that when trained from scratch with adding an adapter (STMA), our adapter module may add complexity to the model, and lead the optimization to more challenging settings. The second question could be whether the two-stage training process is simply a supplementary tool for using the adapter. While it could be argued that the two-stage approach is simply for incorporating the adapter, the true effect of our two-stage approach is in the transfer learning associating two different domains: image and video. In Table 1, we verify the effectiveness of two-stage approach without STMA, which already achieves considerable improvement over the baseline model. When comparing the results between 'PATHS' and 'w/o STMA' in two-stage method, STMA also brings further improvement when applied in the two-stage scheme. Table 1 also illustrates the effectiveness of freezing and pivoting, which works complementary and supportive through the two-stage process.

## 4.2 SCENE TRANSITION MASK ADAPTER

We introduce a generally applicable adapter called STMA. This adapter can be inserted as a module into pre-trained weight of CLIP in video retrieval tasks as a transfer learning method, which can be easily applied as a plug-and-play component. Figure 4 illustrates the overall structure of STMA, and shows how the module is inserted into the network. As shown in the figure, STMA is a small module inserted into the temporal encoder. The major functionality of STMA is to aid the model in learning the concept of scenes in an input video. This approach is inspired by a tree-based video segmentation algorithm proposed in (Kang et al., 2023), which compares the similarity between frames for splitting scenes. In our proposed method, a mask that can divide scenes in the input video is generated using the forward difference of similarity between each video's first frame.

Applying the generated mask to the input video, we can split the input video to multiple scenes, and these scenes subsequently pass through a co-attention layer. Different from the self-attention layer in the transformer structure, the co-attention layer takes different queries, keys, and values to enable the learning and updating of two pieces of information regarding each other. Through this process, STMA leverages the co-attention layer to learn the correlation between the first and second scenes divided within the video. In the following step, the learned first and second scenes are combined with the output of the existing temporal encoder, namely, each block's video representation. This combination again passes through an attention layer to identify the most crucial part of the video throughout the entire video and each scene. Convolution 1d, max pooling, and average pooling reduce the increased dimensions via a pooling layer, which is followed by the Feed Forward Network previously trained in the temporal encoder.

The extra learnable parameters in STMA are represented in red blocks in Figure 4, and in fact, our adapter layer only requires approximately 3% more parameters from the existing model. Besides, STMA is deactivated in the first stage model and is additionally trained after completely freezing the existing model except STMA in the second stage. A detailed description of STMA can be found in Appendix A.3.

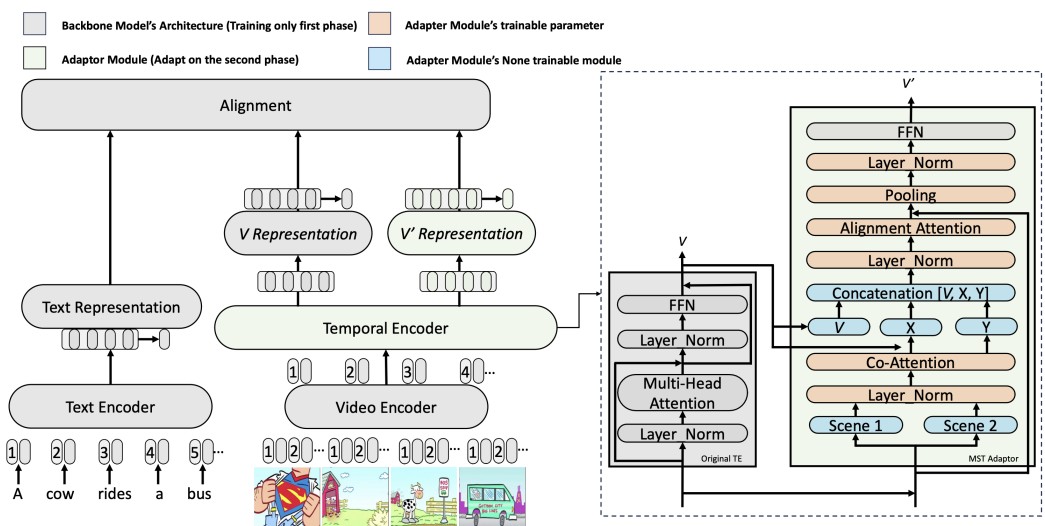

Figure 4: The overview of Scene-Transition Mask Adapter (STMA), a key adapter module within the PATHS framework. STMA is a model-agnostic adapter which can be applied to any text-video multimodal architecture through implemented in the temporal encoder. The gray blocks (both in and out of the 'Temporal Encoder') are trained only in the first stage and freezed in the second stage. During the second stage, our STMA, depicted in green blocks, is activated, and the parameters of the red blocks are subsequently learned. The alignment attention utilizes the same encoder as co-attention, trained for scene-specific knowledge (see Appendix A.3 for details).

## 5 EXPERIMENTS

### 5.1 EXPERIMENTAL DETAILS

We perform experimentation on widely used benchmark datasets for text–video retrieval, namely LSMDC, MSRVTT, and MSVD. PATHS has been applied to existing strong baseline models such as CLIP4Clip, TS2-Net, X-CLIP, EMCL and DiCoSA. We present the efficacy of PATHS by comparing the results of each model with PATH to the results of each original model. Throughout the experiments, released codes from each baseline are used, where we implement PATHS directly on the baseline codes.

We use two evaluation metrics to evaluate the performances: recall at rank $K$ (R@K) and mean rank (MnR). The R@K metric measures the fraction of query pairs that fall within the top $K$ results retrieved, where a higher value indicates better performance. MnR signifies the average of matched pairs within the retrieval ranking order, and a lower score in this metric is considered more favorable.

Our model employed a text and video encoder using CLIP (ViT-B/32, ViT-B/16). We use four Nvidia A100 80GB GPUs, starting with a learning rate of 1e-4. With a model batch size of 128 and a validation batch size of 32, we evaluate five baseline models, contrasting their performance against the results achieved with our approach[1], which is provided in Section 5.3.

### 5.2 DATASETS

**LSMDC** dataset contains 118,081 clips extracted from 112 films, with durations between 2 and 30 seconds. Every video is coupled with one caption. We follow the official split: 101,079 for training, 7,408 for validation, and 1,000 for testing.

**MSRVTT** dataset contains 10,000 videos with durations ranging from 10 to 30 seconds. Each video clip is associated with 20 sentences. We follow the same split from previous studies.

---

[1]All experiments with baselines were performed following the original setting and using the authors' original codes in our environment. Consequently, some results might differ from those in the original publications.

**MSVD** dataset contains 1,970 videos with durations ranging from 1 to 62 seconds. On average, each video clip is associated with 40 sentences. We follow the official split.

| Methods | LSMDC | | | | MSRVTT | | | |
|---|---|---|---|---|---|---|---|---|
| | R@1↑ | R@5↑ | R@10↑ | MeanR↓ | R@1↑ | R@5↑ | R@10↑ | MeanR↓ |
| ViT-B-32 | | | | | | | | |
| CenterCLIP (Zhao et al., 2022)SIGIR'22 | 21.9 | 41.1 | 50.7 | 57.2 | 44.2 | 71.6 | 82.1 | 15.1 |
| DRL (Wang et al., 2022)arXiv'22 | 24.9 | 45.7 | 55.3 | - | 47.4 | 74.6 | 83.8 | 12.8 |
| QB-Norm (Bogolin et al., 2022)CVPR'22 | 17.8 | 37.7 | 47.6 | - | 47.2 | 73.0 | 83.0 | - |
| *X-Pool (Gorti et al., 2022)CVPR'22 | 21.5 | 41.6 | 49.8 | 59.2 | 46.8 | 73.3 | 82.4 | 13.9 |
| STAN (Liu et al., 2023)CVPR'23 | 23.7 | 42.7 | 51.8 | - | 46.9 | 72.8 | 82.8 | - |
| HBI (Jin et al., 2023a)CVPR'23 | - | - | - | - | 48.6 | 74.6 | 83.4 | - |
| MEME (Kang & Cho, 2023)SIGIR'23 | 24.0 | 41.7 | 51.4 | 53.5 | 49.0 | 73.5 | 82.0 | 13.0 |
| *CLIP4Clip (Luo et al., 2022)Neurocomputing'22 | 22.8 | 40.3 | 48.6 | 62.3 | 43.0 | 71.9 | 81.8 | 15.6 |
| *CLIP4Clip (+PATHS) | 23.7(+0.9) | 41.8(+1.5) | 49.9(+1.3) | 56.8(-5.5) | 44.5(+1.5) | 72.2(+0.3) | 82.6(+0.8) | 14.6(-1.0) |
| *TS2-Net (Liu et al., 2022)ECCV'22 | 20.8 | 40.8 | 47.8 | 68.1 | 46.6 | 74.9 | 82.5 | 14.1 |
| *TS2-Net (+PATHS) | 22.3(+1.5) | 39.7(-1.1) | 49.2(+1.4) | 66.0(-2.1) | 47.8(+1.2) | 74.1(-0.8) | 82.6(+0.1) | 13.5(-0.6) |
| *EMCL-Net (Jin et al., 2022)NeurIPS'22 | 22.1 | 43.4 | 52.6 | 57.0 | 46.8 | 74.2 | 82.4 | 13.2 |
| *EMCL-Net (+PATHS) | 24.2(+2.1) | 43.9(+0.5) | 52.7(+0.1) | 57.0(±0.0) | 48.3(+1.5) | 74.2(±0.0) | 82.5(+0.1) | 13.0(-0.2) |
| *X-CLIP (Ma et al., 2022)ACMMM'22 | 23.3 | 42.4 | 50.9 | 55.6 | 47.2 | 73.5 | 82.5 | 13.8 |
| *X-CLIP (+PATHS) | 25.1(+1.8) | 42.9(+0.5) | 52.3(+1.4) | 54.9(-0.7) | 48.4(+1.2) | 73.7(+0.2) | 82.7(+0.2) | 13.2(-0.6) |
| *DiCoSA (Jin et al., 2023b)IJCAI'23 | 22.5 | 43.6 | 52.6 | 55.6 | 47.6 | 74.4 | 83.4 | 13.2 |
| *DiCoSA (+PATHS) | 24.1(+1.6) | 43.9(+0.3) | 52.8(+0.2) | 55.4(-0.2) | 49.4(+1.8) | 74.5(+0.1) | 83.7(+0.3) | 12.8(-0.4) |
| ViT-B-16 | | | | | | | | |
| CenterCLIP (Zhao et al., 2022)SIGIR'22 | 24.2 | 46.2 | 55.9 | 47.3 | 48.4 | 73.8 | 82.10 | 13.8 |
| DRL (Wang et al., 2022)arXiv'22 | 26.5 | 47.6 | 56.8 | - | 50.2 | 76.5 | 84.7 | 12.4 |
| STAN (Liu et al., 2023)CVPR'23 | 27.1 | 49.3 | 58.7 | - | 50.0 | 75.2 | 84.1 | - |
| *CLIP4Clip (Luo et al., 2022)Neurocomputing'22 | 25.6 | 45.6 | 55.3 | 53.7 | 46.5 | 73.9 | 82.3 | 15.3 |
| *CLIP4Clip (+PATHS) | 26.1(+0.5) | 46.4(+0.8) | 55.8(+0.5) | 50.4(-3.3) | 47.8(+1.3) | 74.5(+0.7) | 82.4(+0.1) | 14.5(-0.8) |
| *TS2-Net (Liu et al., 2022)ECCV'22 | 20.9 | 42.6 | 51.4 | 63.7 | 48.5 | 75.6 | 84.6 | 13.5 |
| *TS2-Net (+PATHS) | 23.2(+2.3) | 41.2(-1.4) | 51.5(+0.1) | 62.4(-1.3) | 49.4(+0.9) | 76.5(+0.9) | 84.7(+0.1) | 12.9(-0.6) |
| *EMCL-Net (Jin et al., 2022)NeurIPS'22 | 27.3 | 45.3 | 54.4 | 51.3 | 49.0 | 76.1 | 83.7 | 12.3 |
| *EMCL-Net (+PATHS) | 28.5(+1.2) | 46.7(+1.4) | 54.7(+1.6) | 49.8(-1.5) | 50.2(+1.2) | 75.7(-0.4) | 84.3(+0.6) | 11.3(-1.0) |
| *X-CLIP (Ma et al., 2022)ACMMM'22 | 25.9 | 46.3 | 56.2 | 51.4 | 49.4 | 75.6 | 84.5 | 12.7 |
| *X-CLIP (+PATHS) | 27.7(+1.8) | 47.1(+0.7) | 56.7(+0.5) | 49.0(-2.4) | 50.6(+1.2) | 76.2(+0.6) | 84.2(-0.3) | 12.4(-0.3) |
| *DiCoSA (Jin et al., 2023b)IJCAI'23 | 26.4 | 45.6 | 54.9 | 50.8 | 48.9 | 75.0 | 82.9 | 13.7 |
| *DiCoSA (+PATHS) | 27.3(+0.9) | 46.4(+0.8) | 55.0(+0.1) | 49.2(-1.6) | 50.5(+1.6) | 76.2(+1.2) | 84.2(+1.3) | 12.8(-0.9) |

Table 2: Results on LSMDC and MSRVTT, ∗ means results are reproduced using the released code.

| Methods | MSVD | | | | | | | |
|---|---|---|---|---|---|---|---|---|
| | R@1↑ | R@5↑ | R@10↑ | MeanR↓ | R@1↑ | R@5↑ | R@10↑ | MeanR↓ |
| | ViT-B-32 | | | | ViT-B-16 | | | |
| CenterCLIP | 47.6 | 77.5 | 86.0 | 9.8 | 50.6 | 80.3 | 88.4 | 8.4 |
| DRL | 48.3 | 79.1 | 87.3 | - | 50.0 | 81.5 | 89.5 | - |
| QB-Norm | 47.6 | 77.6 | 86.1 | - | - | - | - | - |
| STAN | 47.5 | 77.6 | 86.5 | - | 51.5 | 80.4 | 88.5 | - |
| MEME | 46.6 | 76.5 | 85.0 | 10.2 | - | - | - | - |
| *CLIP4Clip | 45.6 | 75.7 | 84.2 | 10.3 | 47.6 | 75.9 | 86.9 | 9.3 |
| *CLIP4Clip (+PATHS) | 46.7(+1.1) | 76.5(+0.8) | 84.9(+0.7) | 9.9(-0.4) | 48.4(+0.8) | 78.9(+3.0) | 87.2(+0.3) | 8.9(-0.4) |
| *X-CLIP | 46.2 | 76.1 | 84.8 | 9.9 | 48.2 | 79.1 | 87.1 | 8.9 |
| *X-CLIP (+PATHS) | 47.2(+1.0) | 76.8(+0.7) | 85.2(+0.4) | 9.6(-0.3) | 49.6(+1.4) | 79.3(+0.2) | 87.6(+0.5) | 8.5(-0.4) |

Table 3: Results on MSVD, ∗ means the results are reproduced using the publicly released code.

## 5.3 EXPERIMENTAL RESULTS

In this section, we compare the performance of the PATHS framework against strong baseline models across various datasets. PATHS, designed for harnessing the pre-trained weights of CLIP for video tasks, is a plug-and-play scheme that can be applied to any existing models. Table 2 and Table 3 show that PATHS always improves existing baselines by a significant margin. It consistently improves performance across all baselines on every metric, notably in R@1 and MeanR. R@1 examines solely the top prediction, which is the in the rigorous setting among R@K. MeanR offers a holistic evaluation by averaging the rankings of all valid annotations.

We achieve SOTA performances in LSMDC and MSRVTT. For LSMDC, the best performance with ViT-B-32 is achieved when PATHS is applied to X-CLIP; the best performance with ViT-B-16 is achieved when PATHS is applied to EMCL. For MSRVTT, the best performance with ViT-B-32 is achieved when PATHS is applied to DiCoSa; the best performance with ViT-B-16 is achieved when PATHS is applied to X-CLIP. Regarding the MSVD dataset[2], applications to the CLIP4Clip and X-

---

[2]Ts2Net, EMCL, and DiCoSA are left out due to the absence of codes from the authors.

CLIP showed a consistent performance surge of over 1%. These results underscore the significance of proposed scheme which further improves all strong benchmark models across all datasets.

## 5.4 ABLATION STUDY

| Strategy | CLIP4Clip | | | TS2Net | | | X-CLIP | | | DiCoSA | | | EMCL-Net | | |
|---|---|---|---|---|---|---|---|---|---|---|---|---|---|---|---|
| | R@1 | R@5 | MeanR | R@1 | R@5 | MeanR | R@1 | R@5 | MeanR | R@1 | R@5 | MeanR | R@1 | R@5 | MeanR |
| per Epoch | 43.0 | 71.9 | 15.6 | 46.6 | 74.9 | 14.1 | 47.2 | 73.5 | 13.8 | 47.6 | 74.4 | 13.2 | 46.8 | 74.2 | 13.2 |
| per 50 Step | 43.9 | 71.3 | 15.8 | 46.9 | 73.2 | 13.8 | 47.9 | 73.6 | 14.7 | 47.9 | 74.8 | 13.6 | 47.2 | 74.3 | 13.2 |
| First Stage | | | | | | | | | | | | | | | |
| BP | 43.7 | 71.6 | 15.7 | 46.8 | 73.3 | 14.7 | 47.8 | 73.5 | 13.6 | 47.6 | 74.4 | 13.2 | 46.8 | 74.0 | 13.3 |
| SP | 43.6 | 70.9 | 15.3 | 46.6 | 75.0 | 14.1 | 47.2 | 73.5 | 13.8 | 47.7 | 74.2 | 13.1 | 47.0 | 74.1 | 13.1 |
| USP | 43.6 | 71.9 | 15.0 | 46.7 | 74.6 | 14.0 | 47.9 | 74.0 | 14.0 | 48.1 | 74.0 | 13.4 | 46.9 | 74.4 | 13.1 |
| Second Stage | | | | | | | | | | | | | | | |
| BP | 43.9 | 71.5 | 15.3 | 48.1 | 74.3 | 13.8 | **48.4** | 73.5 | 13.3 | 48.9 | 74.8 | 12.5 | 48.1 | 74.0 | **12.8** |
| SP | **45.0** | 72.0 | 14.7 | **48.2** | **74.5** | **13.4** | **48.4** | 73.2 | 13.3 | 49.0 | **74.9** | 12.4 | 48.1 | 74.0 | 13.0 |
| USP | 44.5 | **72.2** | **14.6** | 47.8 | 74.1 | 13.5 | **48.4** | **73.7** | **13.2** | **49.4** | 74.5 | 12.8 | **48.3** | **74.2** | 13.0 |

Table 4: Ablation study on measuring dynamics for PATHS. MSRVTT dataset has been used.

We conduct comprehensive ablation experiments to analyze how each component of our model contributes to the performance. Additional ablation studies focusing on adapter configurations are delineated in Appendix A.4 (see Tables 6,7). In Table 4, we first juxtapose the traditional epoch-based evaluation paradigm with an alternative scheme that assesses model performance at fixed intervals of $N$ steps. Empirical results indicate that the model performance with evaluations at every 50-step improves their epoch-based counterparts. However, frequent evaluations incur computational overhead, which we further discuss in Appendix A.1.

We investigate variations of quantifying dynamics for PATHS. The first strategy denoted as **BP** in Table 4 targets at finding the $K$ most salient fluctuations in ranking of parameters within a group. The second strategy denoted as **SP** monitors the parameters with the longest stability, and captures the moment when the parameters become unstable. This is achieved by monitoring instances where parameter ranking shifts remain below the standard deviation, thereby indicating model stability. Our final strategy denoted as **USP** is the opposite of the second strategy. We monitor the parameters which exhibit the longest unstableness, and capture the moment when the parameters become stable. This approach identifies instances where parameter ranking fluctuations surpass the moving average of previously observed rank changes. We found USP to be most effective in our experiments, and all the reported results in Tables 2 and 3 are obtained with USP.

A preliminary examination of Table 4 reveals that, even at the first stage, the PATHS framework is on par with the performance achieved by 50-step interval evaluations, while offering significant computational efficiencies. Extending our methodology by incorporating a second-stage adapter refinement, we demonstrate that PATHS consistently outperforms existing models evaluated at recurring 50-step intervals across various performance metrics.

## 6 CONCLUSION

In this paper, we discuss the limitation of current text-to-video retrieval models built upon CLIP, where overfitting often occurs during fine-tuning over the training data. To tackle the problem, we propose Parameter-wise Adaptive Two-stage training Harnessing Scene transition mask adapter (PATHS). PATHS is a generic framework which is applicable to any framework built upon CLIP. Numerical results from extensive experiments support the efficacy of PATHS. PATHS always achieves significant improvements on strong baselines when applied in a plug-and-play manner. On LSMDC and MSRVTT data, we achieve SOTA results in every aspect. The ablation study further illustrates the effectiveness of each component of PATHS.

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

# A    APPENDIX

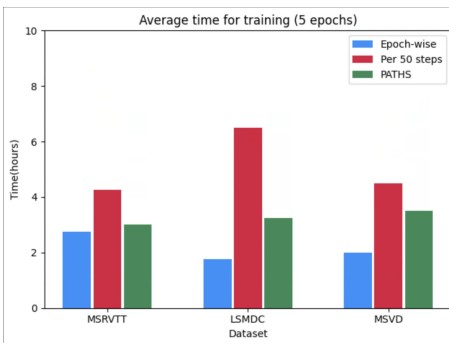

Figure 5: The full training time in hours are compared. For training, each evaluation scheme has been adopted. Each bar is an average time of all models used in the experimentations.

| Methods | Accuracy↑ |
|---|---|
| ClipBERT (Lei et al., 2021) | 37.4 |
| VGT (Xiao et al., 2022) | 39.7 |
| VQA-T (Yang et al., 2021) | 41.5 |
| SiaSamRea (Yu et al., 2021) | 41.6 |
| MERLOT (Zellers et al., 2021) | 43.1 |
| Co-Tok (Piergiovanni et al., 2022) | 45.7 |
| *EMCL (Jin et al., 2022) | 45.2 |
| *EMCL (+PATHS) | **45.5(+0.3)** |

Table 5: Performance comparison of video question answering on MSRVTT. PATHS can also be applied to EMCL for the task, which achieves performance improvement. We use original code of EMCL for the experimental evaluation.

## A.1    TIME EFFICIENCY

In Figure 5, we present a comparative analysis of time complexities across three evaluation paradigms: traditional epoch-wise evaluation, $N$-step evaluation, and the PATHS evaluation method. In text-to-video retrieval literature, all models are trained over 5 epochs, and we follow the protocol reporting the time for training over 5 epochs. Figure 5 is the average total training time in hour of the previous models reported in Table 2, and 3. The number of videos and, the number of frames of each video have strong differences across three datasets, which is reflected in the bar graph in Figure 5.

Notably, $N$-step evaluation requires significant increase in time for training each of the datasets. This is mainly because of the frequent evaluations on validation set at every epoch, which causes computational overhead. While many recent studies take the $N$-step approach, less attention has been paid on the computational overhead problem. Our study addresses this problem proposing a novel approach which doesn't require frequent evaluations. For MSRVTT dataset, PATHS only takes approximately the same amount of time for training as the conventional epoch-wise method. For both LSMDC and MSVD datasets, PATH saves the training time compared to the 50-step approach. When number of video gets large, such as the LSMDC dataset, the improvement becomes significant. It is also worth to note that PATHS performs as nearly as the $N$-step evaluation when compared one-to-one (see per 50 step vs first stage in Table 4).

## A.2    PATHS APPLIED TO VIDEO QUESTION ANSWERING TASK

We further investigate how PATHS can be applied to a task other than text-to-video retrieval. Several works have been proposed in video question answering (VideoQA) task. VideoQA leverages visual information from videos to predict corresponding answers of input questions. Utilizing the target vocabulary designed for the MSRVTT-QA dataset (Xu et al., 2016), we trained a fully connected layer atop the final linguistic features to categorize the answer.

In Table 5, EMCL (Jin et al., 2022) is the only model that leverages the pre-trained weights from CLIP. We use EMCL as a baseline for the given task, and see how PATH can further improve EMCL in VideoQA. From Table 5, we confirm that PATHS can also improves EMCL on VideoQA task, which highlights its generalization to other tasks.

## A.3    THE PROCESS OF THE STMA

This section elaborates the mechanics STMA, which is conceptualized in Figure 6. The STMA begins by bisecting a video based on a scene transition algorithm, encapsulated within the green

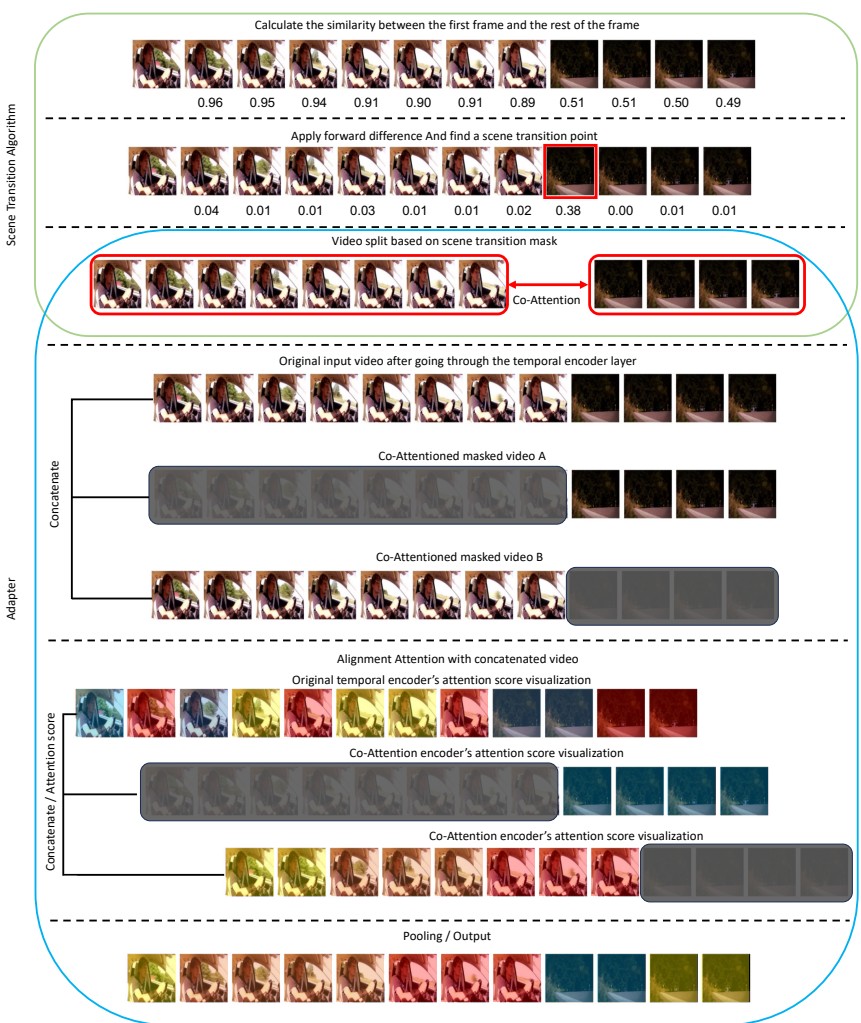

Figure 6: The figure presented serves as a schematic representation of our STMA. The segment enclosed within the green box illustrates the algorithm's mechanism for identifying scene transition points. The blue box emphasizes the functional deployment of STMA within the temporal encoder. The color presents the attention score: red signifies high attention scores, yellow indicates medium levels, and blue represents low scores.

box in the figure. This algorithm initiates by assessing the similarity between the initial frame and subsequent frames as detailed in Section 4.2. We then identify the scene transition point by the similarity matrix. After that, we create a mask before and after the corresponding scene transition point.

The blue box in the figure outlines the process of STMA. This module ingests two masks generated from the scene transition algorithm, and processes three frame sequences within the model. The first sequence is the original video post-attention layer, while the remaining sequences are mask-modified frames subjected to co-attention mechanisms. The multiple processes allows the model to capture the multiple aspects of scenes within a given video. Finally, we apply additional attention layer and pooling layer for obtaining final output which reflects the importance among the three processes.

Our model internalizes knowledge of scenes originating from a singular video source through the co-attention module. We use a single encoder so that the alignment attention module and co-attention

module share the same encoder. This approach can minimize the additional parameters, and also turn out to be more effective in performance than having two separate encoder. In general, training co-attention and multi-head attention simultaneously in the same encoder hinders effective optimizations. However, in this unique scenario, where each of the frames from a same video exhibit strong correlations, the problem is mitigated. We can additionally mitigate the problem with our training strategy. In the second stage of our training scheme, all irrelevant parameters are kept frozen, and thus attention weights can be updated effectively.

## A.4 ABLATION STUDY FOR STMA

| Gate | Text-to-Video Retrieval | | | |
|---|---|---|---|---|
| | R@1 | R@5 | R@10 | MeanR |
| Baseline | 47.2 | 73.5 | 82.5 | 13.8 |
| w/o CoAttn | 47.8 | 73.3 | 81.8 | 13.6 |
| w/o A-Attn | 47.8 | 73.4 | 82.2 | 13.7 |
| Full STMA | **48.4** | **73.7** | **82.7** | **13.2** |

Table 6: Ablation study of attention mechanisms on STMA. Text-to-video retrieval results on MSRVTT are compared. We use X-CLIP as a baseline.

| Gate | Text-to-Video Retrieval | | | |
|---|---|---|---|---|
| | R@1 | R@5 | R@10 | MeanR |
| Baseline | 47.2 | 73.5 | 82.5 | 13.8 |
| Avg | 48.1 | 73.6 | **82.9** | 13.4 |
| Max | 48.1 | 73.4 | **82.9** | **13.2** |
| Conv | **48.4** | **73.7** | 82.7 | **13.2** |

Table 7: Ablation study of pooling strategies on STMA. Text-to-video retrieval results on MSRVTT are compared. We use X-CLIP as a baseline.

To understand how the co-attention and alignment attention contribute to the STMA performance, we perform ablative analysis, where the results are provided in Table 6. We confirm that each component indeed contributes to the STMA performance. The full implementation of STMA brings significant performance improvements. More importantly, in full implementation, each component contributes complementarily to the final results.

Additionally, Table 7 illustrates the performance with different pooling techniques employed in the STMA. We compare average pooling, max pooling, and convolution-1D pooling methods. Among these, the convolution-1D approach demonstrated the best results.

## A.5 VIDEO-TO-TEXT RETRIEVAL

| Methods | MSRVTT | | | | | | | |
|---|---|---|---|---|---|---|---|---|
| | R@1↑ | R@5↑ | R@10↑ | MeanR↓ | R@1↑ | R@5↑ | R@10↑ | MeanR↓ |
| | ViT-B-32 | | | | ViT-B-16 | | | |
| CenterCLIP (Zhao et al., 2022)SIGIR'22 | 45.1 | 72.4 | 83.1 | 10.0 | 47.7 | 75.0 | 83.3 | 10.2 |
| DRL (Wang et al., 2022)arXiv'22 | 45.3 | 73.9 | 83.3 | - | 48.9 | 76.3 | 85.4 | - |
| *X-Pool (Gorti et al., 2022)CVPR'22 | 44.4 | 73.3 | 84.0 | 9.0 | - | - | - | - |
| MEME (Kang & Cho, 2023)SIGIR'23 | 47.7 | 74.0 | 83.3 | 9.4 | - | - | - | - |
| *CLIP4Clip (Luo et al., 2022)Neurocomputing'22 | 43.2 | 70.5 | 80.2 | 11.8 | 45.7 | 72.4 | **83.2** | 10.8 |
| *CLIP4Clip (+PATHS) | 43.4(+0.2) | 71.5(+1.0) | 81.4(+1.2) | 10.8(+1.0) | 46.3(+0.6) | 75.3(+2.9) | 83.0(-0.2) | 9.6(-1.2) |
| *TS2-Net (Liu et al., 2022)ECCV'22 | **46.1** | 73.8 | 83.5 | 9.4 | 46.8 | 76.7 | 84.8 | 8.6 |
| *TS2-Net (+PATHS) | 45.9(-0.2) | 74.0(+0.2) | 84.4(+0.9) | 8.9(-0.5) | 47.9(+1.1) | 77.4(+0.7) | 86.6(+1.8) | 8.2(-0.4)) |
| *EMCL-Net (Jin et al., 2022)NeurIPS'22 | 46.1 | 73.7 | **84.2** | 9.8 | 50.2 | 75.7 | 84.0 | 8.8 |
| *EMCL-Net (+PATHS) | 48.0(+1.9) | 74.8(+1.1) | 83.8(-0.4) | 9.2(-0.6) | 51.1(+0.9) | 77.0(+1.3) | 85.1(+1.1) | 8.3(-0.5) |
| *X-CLIP (Ma et al., 2022)ACMMM'22 | 47.2 | 73.2 | 80.6 | 10.5 | 47.6 | **77.3** | 84.8 | 8.8 |
| *X-CLIP (+PATHS) | 47.7(+0.5) | 73.6(+0.4) | 82.3(+1.7) | 9.5(-1.0) | 48.8(+1.2) | 75.9(-1.4) | 84.9(+0.1) | 8.6(-0.2) |
| *DiCoSA (Jin et al., 2023b)IJCAI'23 | **47.6** | 74.2 | 83.7 | 8.8 | 49.9 | **77.8** | 85.3 | 8.5 |
| *DiCoSA (+PATHS) | 47.2(-0.4) | 74.9(+0.7) | 83.9(+0.2) | 8.7(-0.1) | 50.2(+0.3) | 77.1(-0.6) | 86.7(+1.4) | 7.9(-0.6) |

Table 8: Video-to-text retrieval task results on MSRVTT, ∗ denotes that the results are reproduced using the publicly released code.

Following previous studies in text-to-video retrieval, we perform video-to-text retrieval task. We report the performance in Table 8, where we use MSRVTT dataset for the evaluation.The experimental setup follows the setups from the existing backbone models. We confirm that PATHS is always effective when applied to any strong baseline models for video-to-text retrieval tasks.

## A.6 QUALITATIVE ANALYSIS

To gain insights into the attention weights assigned by STMA and to evaluate its efficacy in scene recognition, we perform qualitative analysis. In the associated figures, attention weights are ex-

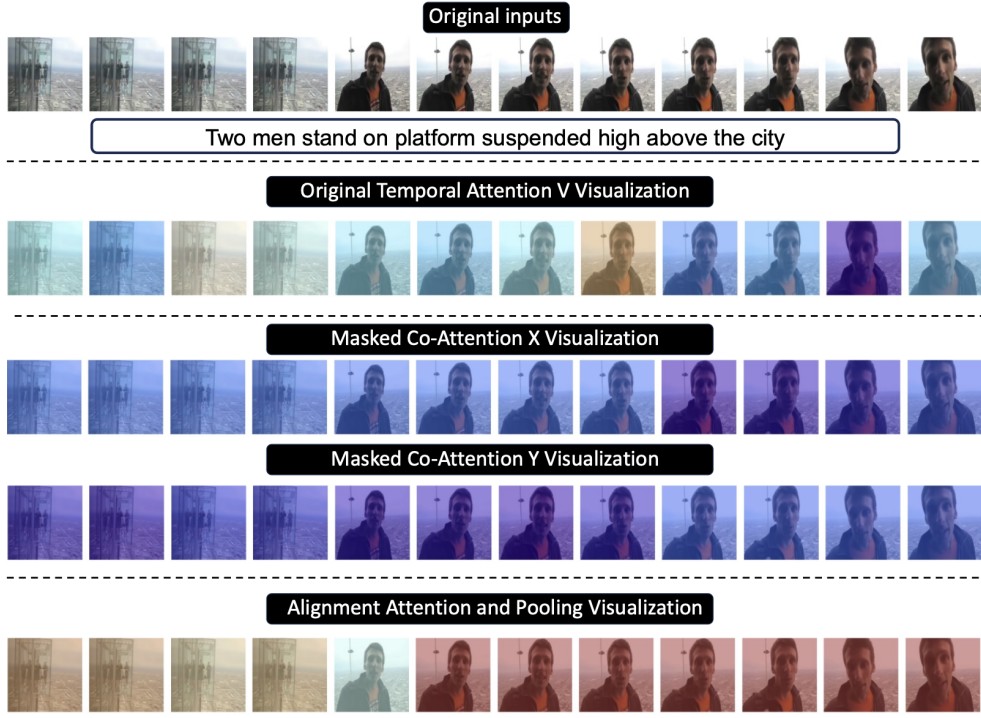

Figure 7: Visualization of the attention score extracted at each part of SMTA.

pressed with colors; red signifies high attention, yellow indicates medium attention, and blue represents low attention. The images in Figure 7, 8, 9, and 10 depict actual experimental results. Distinct from the visualizations generated by original temporal attention, the final output from STMA clearly identifies a sequence of scenes around the transition point. This substantiates that our model is proficient in contextual scene learning.

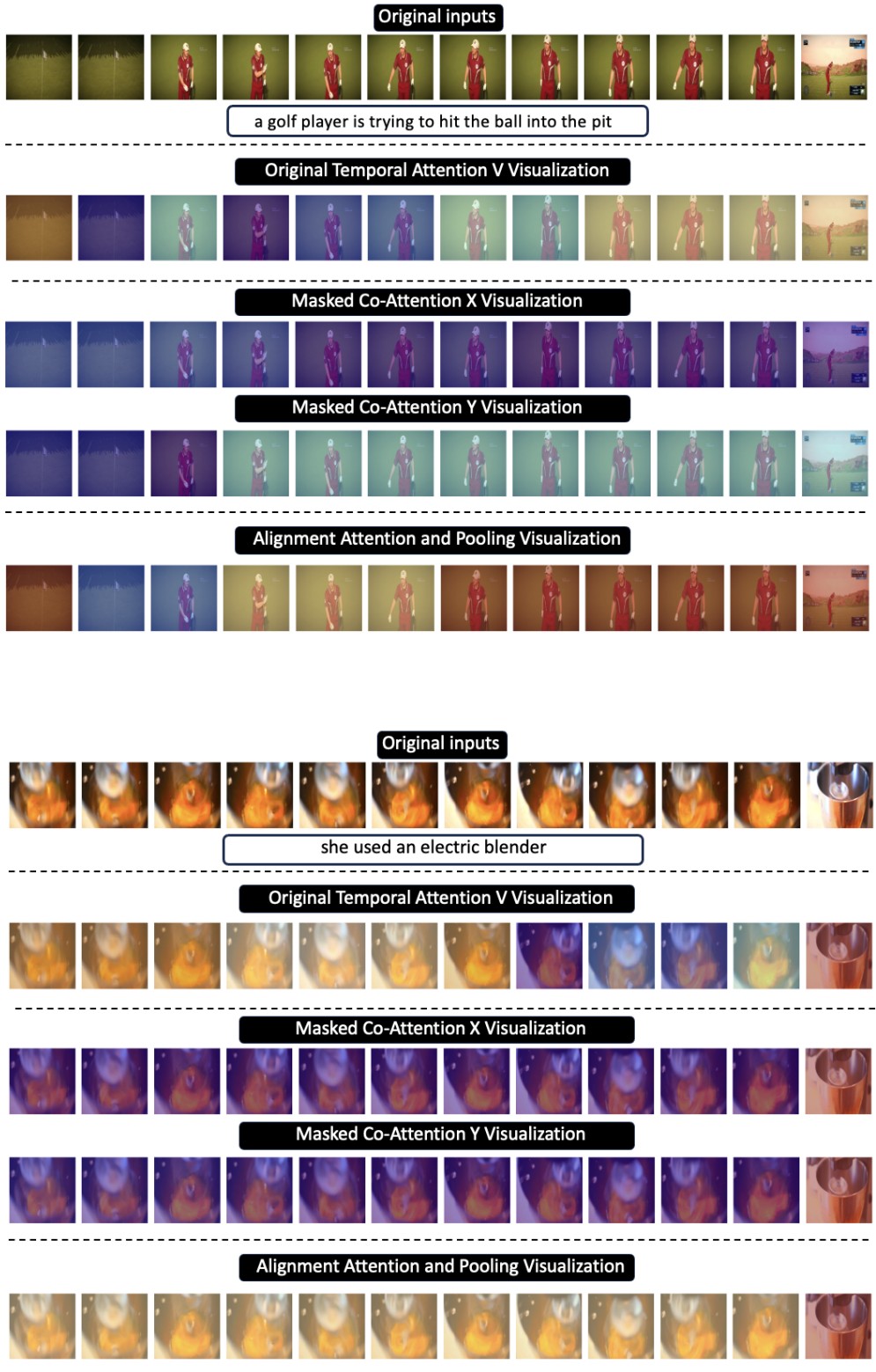

Figure 8: Visualization of the attention score extracted at each part of SMTA.

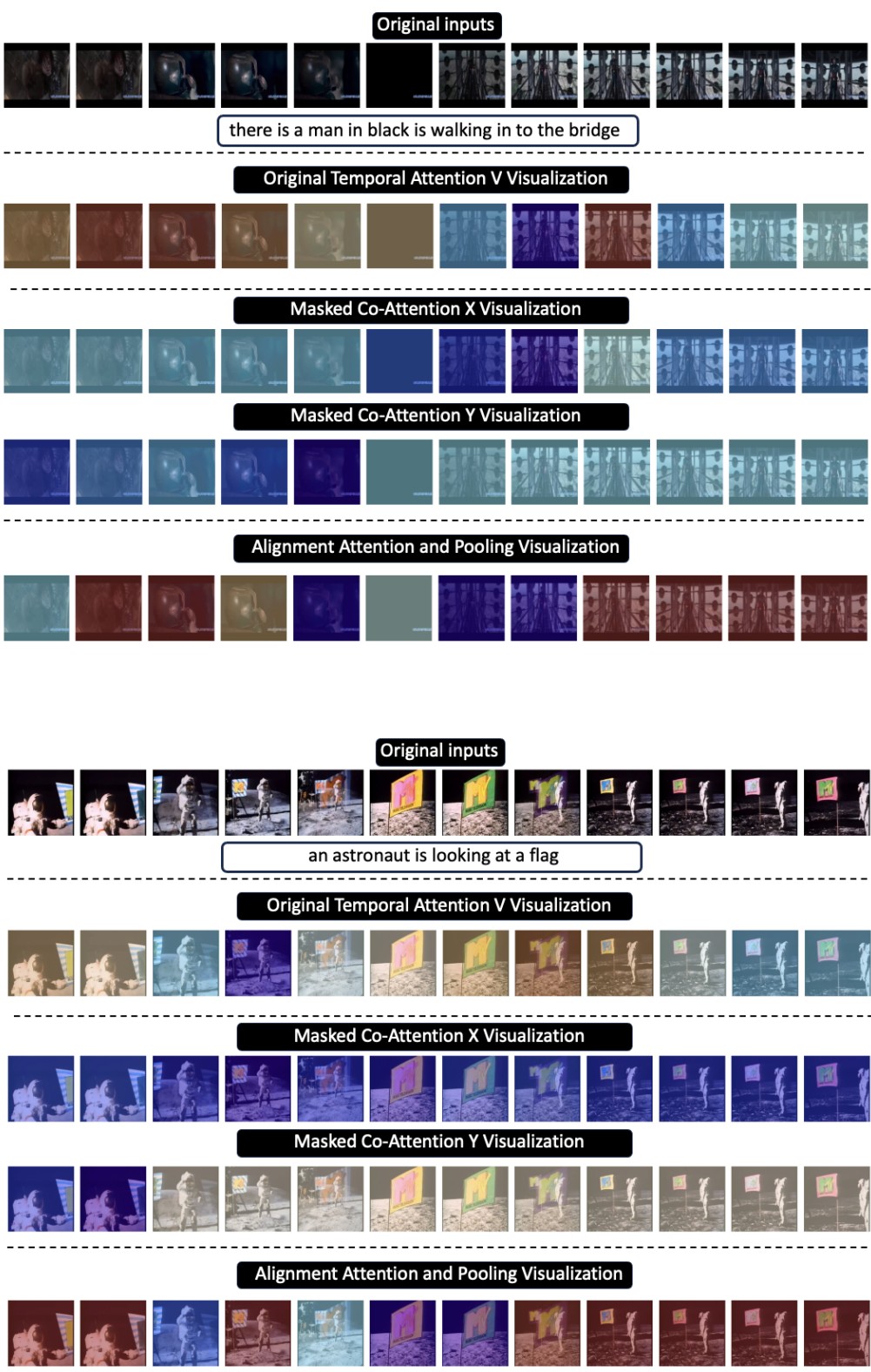

Figure 9: Visualization of the attention score extracted at each part of SMTA.

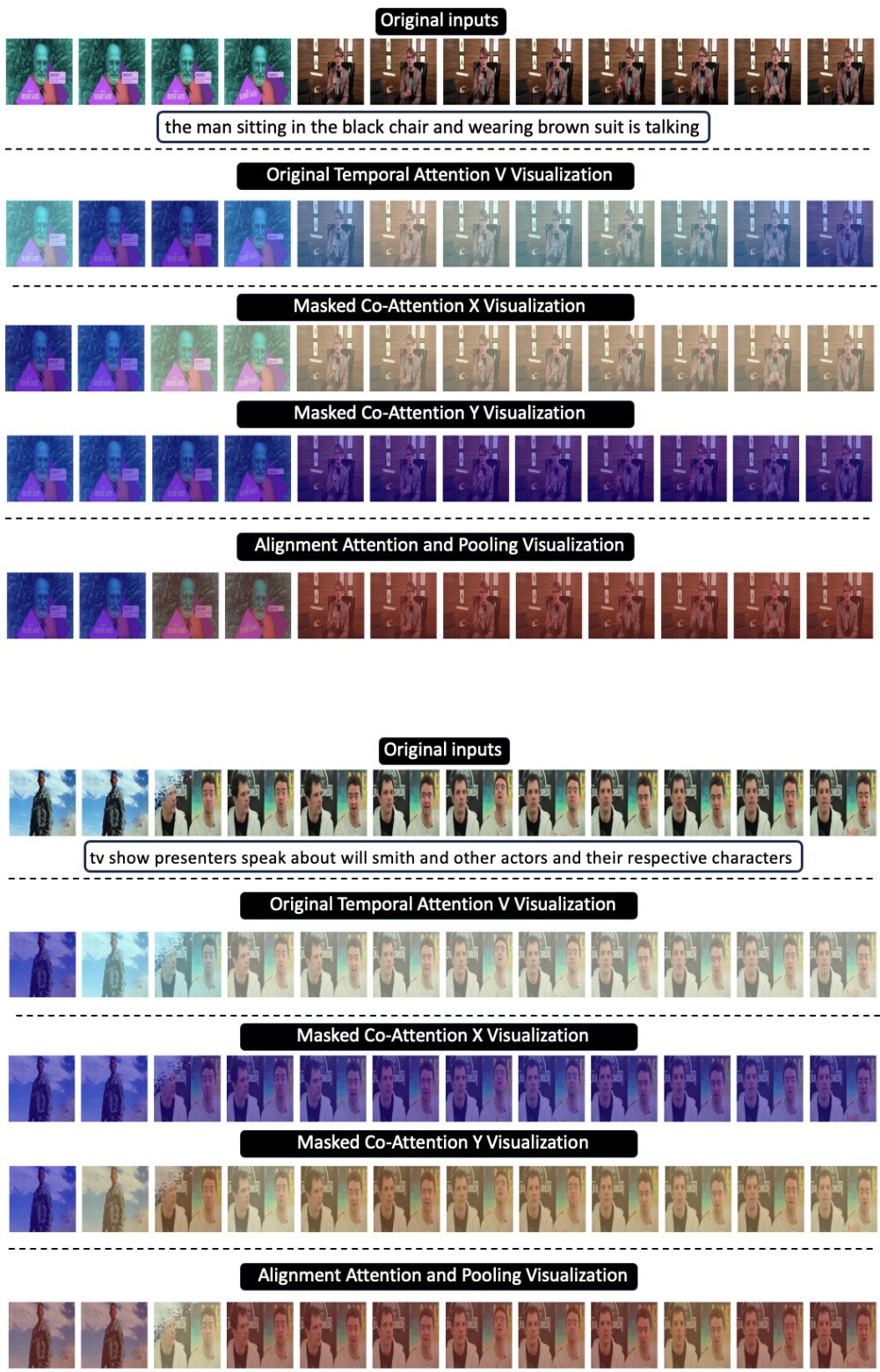

Figure 10: Visualization of the attention score extracted at each part of SMTA.

