# OpenReview forum: "PATHS: Parameter-wise Adaptive Two-Stage Training Harnessing Scene Transition Mask Adapters for Video Retrieval"
_ICLR.cc/2024/Conference — Submitted to ICLR 2024_

### Official Review · Reviewer_hfBT · 2023-10-30

**Soundness:** 4 excellent
**Presentation:** 4 excellent
**Contribution:** 3 good
**Rating:** 8
**Confidence:** 3

**Summary:**

This paper propose a 2 stage method “PATHS” to address the main weakness of the current text-to-video retrieval CLIP-based models, i.e., overfitting. The proposed method includes 2 stages: 1) select best params by monitoring the fluctuations of params, which is much cheaper than e.g. per 50 step eval; 2) train an adapter module STMA with CLIP params frozen.

The proposed method is generic enough to be applicable to any existing CLIP-based models. The result achieves SOTA in the common text-to-video retrieval tasks: LSMDC and MSRVTT.

**Strengths:**

The method is relatively simple and the result is strong with several SOTA.

The experiments are extensive with many baselines; STMA is applied to a wide range of models to show its effectiveness.

The paper is well written and very readable.

The code is open-sourced.

**Weaknesses:**

5.4 ablates the param selection strategies, which is great; also it shows stage-1’s importance in Tab 4. However, it’s still unclear what the performance would be like if we apply stage-2 ONLY to the common param selection strategies (i.e. skip stage-1). This might make it clearer how important stage 1 and 2 is respectively?

**Questions:**

Appendix seems not uploaded?

IIUC, the red curve of Fig 3 is comparable with the green curve of Fig 2 (both are X-CLIP eval), but it seems there’s some difference (e.g. in the end of epoch 5)? If my understanding is correct, maybe it would be clearer if we plot the green curve of Fig 2 in Fig 3 as well?

In 4.1 “Two-stage Process”: in the 2nd stage, if params are frozen, why do we still need to “load these parameters back into the model at the end of each epoch to perform the pivoting”? And could you please explain a bit what “perform the pivoting” exactly means?

mild comments:
typo in 2.2: “adapters have been *unsed* for progressive learning…”

---

> ### Author Response · Authors · 2023-11-13
> **Response to Reviewer hfBT**
>
> First of all, please accept our apologies for submitting the appendix through supplementary material only, which might have been missed as it was not appended to the main draft.
>
> We rectified this error by updating the submission to include the appendix within the manuscript. We are deeply sorry for any inconvenience this has caused and would be grateful if you could take a look at the appendix with reconsideration.
>
> We sincerely thank you for the constructive comments. We address the concerns below.
>
> ---
>
> >5.4 ablates the param selection strategies, which is great; also it shows stage-1’s importance in Tab 4. However, it’s still unclear what the performance would be like if we apply stage-2 ONLY to the common param selection strategies (i.e. skip stage-1). This might make it clearer how important stage 1 and 2 is respectively?
>
> Thank you for your remarks regarding the clarity of our two-stage method's performance and the potential concerns of overfitting.
>
> Regarding the ablation of parameter selection strategies in Section 5.4, we have conducted experiments using X-CLIP and CLIP4Clip as backbones to assess the isolated impact of the second stage without the first (parameter-wise method). Unfortunately, due to space constraints and the structure of Table 4, these results were not included in the manuscript. When the second stage of our method was applied without the parameter-wise initial stage (with initialization done epoch-wise), we observed a performance of Table below. We recognize the importance of this information and will include these details in the appendix of our revised manuscript for a comprehensive understanding.
>
> | Model | R@1 | R@5 | MeanR |
> |-------|--------|-------|--------|
> | X-CLIP | 47.7 | 73.8 | 13.4 |
> |CLIP4Clip| 43.9 | 72.2 | 15.1 |
>
>
> ---
>
> >Appendix seems not uploaded?
>
> Thank you for informing us regarding the appendix. The appendix was initially uploaded only through the supplementary material. We would like to append the appendix at the end of main draft for better readability. Once again, we appreciate your attention to this matter and assistance in improving our submission.
>
> ---
>
> >IIUC, the red curve of Fig 3 is comparable with the green curve of Fig 2 (both are X-CLIP eval), but it seems there’s some difference (e.g. in the end of epoch 5)? If my understanding is correct, maybe it would be clearer if we plot the green curve of Fig 2 in Fig 3 as well?
>
> We appreciate your observation regarding the comparison between the red curve in Figure 3 and the green curve in Figure 2, both of which represent evaluations of the X-CLIP model. The red curve in Figure 3 illustrates the performance at points selected by our parameter-wise method. As such, it does exhibit differences from the green curve in Figure 2, which represents evaluations conducted at fixed intervals (per 50 epochs).
>
> The x-axis in Figure 2 corresponds to these fixed intervals, whereas the x-axis in Figure 3 corresponds to evaluations conducted five times per epoch, with K set to 5. This difference in evaluation frequency makes a direct overlay of the two curves challenging. However, we recognize the need for clearer comparative visualization to facilitate reader understanding. We will explore ways to integrate these two plots more effectively and update the figures accordingly in our revised manuscript.
>
> ---
>
> >In 4.1 “Two-stage Process”: in the 2nd stage, if params are frozen, why do we still need to “load these parameters back into the model at the end of each epoch to perform the pivoting”? And could you please explain a bit what “perform the pivoting” exactly means?
>
> Thank you for inquiring about the process described in Section 4.1 of our manuscript. During the second stage, while the original parameters of the model are indeed frozen, we continue to train the adapter. In this stage, we monitor and pivot the ranking of relative changes within the adapter's internal group of parameters.
>
> Pivoting, in this context, refers to restoring the model's parameters to the state with the highest recorded performance at the end of each epoch. This is achieved by reloading the saved parameters, allowing the model to revert to the optimal point before continuing the training process. This technique helps to fine-tune the model by leveraging the most influential parameter settings discovered during the training epochs.
>
> We will ensure that this explanation is clarified within our manuscript to assist readers in understanding the nuanced training dynamics of our proposed two-stage process.
>
> ---
>
> >mild comments: typo in 2.2: “adapters have been unsed for progressive learning…”
>
> We are deeply thankful for your astute attention to detail concerning the typographical error found in our manuscript. This has been immediately corrected in the text.
>
> ---
>
> Thank you for your time and consideration.
>
> Best regards,

---

> ### Author Response · Authors · 2023-11-21
>
> Dear reviewer hfBT,
>
> We are grateful for your constructive feedback.
>
> We would appreciate informing us whether your concerns have been adequately addressed.
>
> Your insights are highly valued, and we would like to let you know that we would happily answer any further questions if any.
>
> Best regards, The Authors

---

> > ### Comment · Reviewer_hfBT · 2023-11-22
> >
> > Most of my concerns have been addressed. I would like to thank authors for the explanation.

---

### Official Review · Reviewer_zzzT · 2023-11-01

**Soundness:** 2 fair
**Presentation:** 2 fair
**Contribution:** 2 fair
**Rating:** 3
**Confidence:** 4

**Summary:**

In this paper, the author proposes a two-stage training method (PATHS) to improve the results of video-retrieval tasks. In the first stage, it selects 5 candidates according to the parameter ranking. And the results of these candidates are better than those epoch candidates. In the second stage, it uses the previous best checkpoint for initialization and adds an adapter for further fine-tuning. Further experiments demonstrate its effectiveness.

**Strengths:**

The motivation is clear and the method is simple to reproduce.

**Weaknesses:**

- The paper is not well organized.
  - The PRELIMINARIES section seems to be redundant, and the `3.1` and `3.1.1` seem to stage for a single section (subsection).
  - Some details are not clear. For example, how does the ranking work? How does a two-stage scheme w/o STMA work (no adapter and freeze the backbone)? How to split scenes in STMA?
- The method seems to be tricky, which overfits a specific test set.
- The Figures 2 and 3 are in low resolution and hard to read.

**Questions:**

- In Page 5, `Main Idea` part, `In accordance with Figure 1, we denote end of each epoch with dotted line` should be `Figure 2`.
- In Page 5, `Motivation` part, `When the model starts to diverge after passing the optimum point, the model parameter values exhibit strong fluctuations. This often involves rearranging parameters in terms of importance (or the value of parameters)`, how can the author get the conclusion?

---

> ### Author Response · Authors · 2023-11-13
> **Response to Reviewer zzzT**
>
> First of all, please accept our apologies for submitting the appendix through supplementary material only, which might have been missed as it was not appended to the main draft. We rectified this error by updating the submission to include the appendix within the manuscript. We are deeply sorry for any inconvenience this has caused and would be grateful if you could take a look at the appendix with reconsideration.
>
> We sincerely thank you for the constructive comments. We address the concerns below.
>
> ---
>
> >The PRELIMINARIES section seems to be redundant, and the 3.1 and 3.1.1 seem to stage for a single section (subsection).
>
> We acknowledge your concerns regarding the structure of our PRELIMINARIES section and the segmentation between sections 3.1 and 3.1.1. We are considering restructuring these sections into a cohesive single section or revising the PRELIMINARIES section to eliminate redundancy and effectively address your concerns. We are grateful for your precise feedback and will refine our manuscript accordingly.
>
> ---
>
> >Some details are not clear. For example, how does the ranking work? How does a two-stage scheme w/o STMA work? How to split scenes in STMA?
>
> Thank you for highlighting areas in our manuscript where further clarity is needed. Regarding the parameter group ranking, it operates as outlined in Section 4.1: Quantifying Dynamics subsection. We monitor the magnitude of change within each parameter group and compare it to the changes in other groups. In other words, Ranking works by comparing the amount of change in the "parameter group". Based on these observations, we use three methods—$BP$, $SP$, and $USP$—to determine the final evaluation points. This process allows us to assess and rank the parameter groups effectively.
>
> In the case of operating without the Scene Transition Mask Adapter (STMA), our two-stage scheme would not freeze the original parameters but would instead continue additional training. We acknowledge that this was not sufficiently detailed in our initial description and will make the necessary amendments to ensure this is clearly articulated in our revised manuscript.
>
> As for scene splitting within STMA, this is elaborated in Appendix A.3. We calculate the similarity between video frames and identify where the largest difference in similarity occur, using these points to segment the video. This method allows the STMA to process and learn from the inherent structure of the video content effectively. We will incorporate these clarifications into the revised version of our manuscript to ensure these processes are transparent and comprehensible for all readers.
>
> ---
>
> >The method seems to be tricky, which overfits a specific test set.
>
> Thank you for your comment on the potential overfitting of our method to a specific test set. We want to clarify that at no point during the training phase do we utilize the test set. Our methodology strictly follows the experimental protocol. That is to say, we strictly adheres to using validation datasets across all datasets, and the parameters selected for final evaluation are those that yield the highest performance on the validation set. The results reported are derived from assessments conducted on the test dataset using these optimally tuned parameters. This approach ensures that our method is evaluated fairly and without bias towards the test set.
>
> ---
>
> >In Page 5, Main Idea part, In accordance with Figure 1, we denote end of each epoch with dotted line should be Figure 2.
>
> We are truly grateful for your keen observation regarding the typographical error in our manuscript. We have promptly addressed and corrected this in the manuscript.
>
> ---
>
> >In Page 5, Motivation part, When the model starts to diverge after passing the optimum point, the model parameter values exhibit strong fluctuations. This often involves rearranging parameters in terms of importance, how can the author get the conclusion?
>
> We apologize for the confusion, we originally meant to use "fluctuation" to describe "rank fluctuation".
>
> As elaborated in the Motivation section on page 5 of our manuscript, the statement *"When the model starts to diverge after passing the optimum point, the model parameter values exhibit strong fluctuations. This often involves rearranging parameters in terms of importance"* is predicated on the previous sentences on manuscript: *"The solution we proposed is based on our key observation. When the model converges, parameters in the neural network exhibit stable behavior. Here, we focus on the other way, i.e., when the model diverges."*
>
> Our hypothesis is grounded in the observation that parameters in a neural network stabilize as the model converges. Conversely, we posit that significant parameter’s rank fluctuations suggest divergence, which indicates instability. This premise has shaped our development of the PATHS and guided our empirical efforts to substantiate its effectiveness.
>
> Thank you for your time and consideration.
>
> Best regards,

---

> ### Author Response · Authors · 2023-11-21
>
> Dear reviewer zzzT,
>
> We are grateful for your constructive feedback.
>
> We would appreciate informing us whether your concerns have been adequately addressed.
>
> Your insights are highly valued, and we would like to let you know that we would happily answer any further questions if any.
>
> Best regards, The Authors

---

> > ### Comment · Reviewer_zzzT · 2023-11-23
> > **Response to the authors**
> >
> > I appreciate the authors' response, but I maintain my initial assessment and am inclined to reject the paper.
> >
> > In my view, the key concept remains somewhat `tricky`. Moreover, regarding the issue of `overfitting`, I harbor concerns that such a strategy could potentially compromise robustness in practical applications. For instance, if we were to implement this strategy during pretraining with the aim of selecting the optimal checkpoint.
> >
> > As for the manuscript, it necessitates significant polishing and improved presentation:
> >
> > (1) The diagrams suffer from low resolution and their color schemes make them difficult to interpret.
> >
> > (2) The font sizes used within the tables lack consistency; furthermore, references are conspicuously absent in Tables 3 & 4. Vital context-related claims should also be incorporated into the captions.
> >
> > (3) The manuscript's structure should be reorganized to better highlight the authors' methodology. For example, visualizing the BP, SP and USP pipeline would be beneficial for readers unfamiliar with the ranking system. Moreover, Figure 6, which pertains to STMA, would ideally be included within Figure 4 to enhance readability.
> >
> > Looking forward, I hope the authors will invest significant effort in refining their paper. From my perspective, the core idea could be neatly partitioned into two segments: architecture and training. I recommend the use of more engaging visuals in lieu of excessive text to depict the motivation and processing pipeline. Providing more ablation studies on the architecture and training, emphasizing efficiency and effectiveness (i.e., performance vs. training time) would also add value. A demonstration of robustness in pretraining contexts, perhaps added to current efficient CLIP video post-pretraining methods, would contribute further strength to the paper.

---

### Official Review · Reviewer_Cekr · 2023-11-02

**Soundness:** 2 fair
**Presentation:** 1 poor
**Contribution:** 2 fair
**Rating:** 5
**Confidence:** 4

**Summary:**

The paper tries to address the weight corruption problem arising when extending pre-trained CLIP weights to tasks in the video domain.

It proposes a new learning strategy named "Parameter-wise Adaptive Two-stage training Harnessing Scene transition mask adapter" (PATHS), which involves a two-phase learning process. The first phase focuses on determining the optimal weights by monitoring parameter fluctuations, while the second phase concentrates on understanding scenes using an adapter module.

The paper demonstrates the effectiveness of their approach by achieving leading performances on major text-video benchmark datasets such as MSRVTT and LSMDC.

**Strengths:**

1. The proposed method does not require frequent evaluations at every N step, which is distinct from recent approaches that incur extra computational overhead.
2. PATHS can be applied to strong baselines in a plug-and-play manner and has shown consistent performance improvements.

**Weaknesses:**

1. The elaboration in Section 4.2 on the proposed **Co-Attention** module in STMA is not clear enough. The sentence '*the co-attention layer takes different queries, keys, and values to enable the learning and updating of two pieces of information regarding each other*' is confusing. What are the settings of QKV in your Co-Attention? Considering this module is part of the core designs, more formulation or illustration is needed for better understanding.
2. If the **Alignment Attention** in Figure 4 is the so-called *'attention layer' utilized to identify the crucial parts of the video*, the authors should clarify its module name in the paper. Is **Alignment Attention** in STMA a vanilla attention module? How does the attention layer *identify the most crucial part of the video throughout the entire video and each scene*? More explanation and evidence are needed to support this.
3. According to the paper, the authors set the hyperparameter $K$ as 5. How do the authors choose this value? More ablation studies on $K$ should be conducted.
4. What if in some certain samples, there does not exist a scene transition or contains more than one transition? Can the proposed STMA be applied to all possible conditions?

**Questions:**

Please see the Weaknesses mentioned above.

---

> ### Author Response · Authors · 2023-11-13
> **Response to Reviewer Cekr (1/2)**
>
> First of all, please accept our apologies for submitting the appendix through supplementary material only, which might have been missed as it was not appended to the main draft. We rectified this error by updating the submission to include the appendix within the manuscript. We are deeply sorry for any inconvenience this has caused and would be grateful if you could take a look at the appendix with reconsideration.
>
> We sincerely thank you for the constructive comments. We address the concerns below.
>
> ---
>
> >The elaboration in Section 4.2 on the proposed Co-Attention module in STMA is not clear enough. The sentence 'the co-attention layer takes different queries, keys, and values to enable the learning and updating of two pieces of information regarding each other' is confusing. What are the settings of QKV in your Co-Attention?
>
> We appreciate the feedback regarding the description of the Co-Attention module. To clarify, our Co-Attention module is designed to update the values of one modality by taking it as a query and using the keys from another modality. Specifically, when we have two scenes, say $A$ and $B$, scene $A$ serves as the query, and the keys from scene $B$ are utilized to update the values corresponding to scene $B$. In light of your comment, we recognize the need for a more detailed explanation and will update the manuscript to include a clearer description of the QKV settings in our Co-Attention module.
>
> ---
>
> >If the Alignment Attention in Figure 4 is the so-called 'attention layer' utilized to identify the crucial parts of the video, the authors should clarify its module name in the paper. Is Alignment Attention in STMA a vanilla attention module? How does the attention layer identify the most crucial part of the video throughout the entire video and each scene?
>
> Thank you for your inquiry regarding the Alignment Attention. Yes, the Alignment Attention Layer within our system is indeed a vanilla attention module. It functions by taking the original video, denoted as $A$, and the scenes $B$ and $C$, which are derived from $A$ and updated through Co-Attention, as inputs. These inputs are then concatenated. The Alignment Attention operates on this concatenated information to compute additional attention scores across the elements, thereby extracting the most salient parts of the video. This process effectively allows the model to identify and focus on the most informative segments throughout the entire video and within each scene. You can find a figure of this in A.3 in the appendix. We acknowledge the need for a more comprehensive explanation in our manuscript and will enhance the manuscript with additional details to better support the functioning and utility of the Alignment Attention.
>
> ---
>
> >According to the paper, the authors set the hyperparameter $K$ as $5$. How do the authors choose this value? More ablation studies on K should be conducted.
>
> Thank you for your question regarding selecting our study's hyperparameter value set to 5. As mentioned in comment 1 of reviewer 1, We acknowledge the trade-off between the upper bound of performance and the computational overhead that comes with increasing the value of this $K$. We chose $5$ to maintain computational efficiency while capturing a significant portion of the performance gains. We posit that a higher evaluation frequency could further enhance performance in cases with larger training datasets and more learning steps. However, for the scope of our experiments, we found that setting this $K$ to $5$ provided a suitable balance between performance and computational resource management. We appreciate the suggestion for additional ablation studies on this $K$ and will consider this for future research to further elucidate its impact.
>
> The suggested ablation experiments (different K) are currently underway.  We wanted to respond to other questions first in a timely manner, and would like to come back once we get the results.
>
> ---
> Below, we continue our response with the results.
>
> We are very thankful for your constructive suggestion. As indicated in the table, there is a trend that performance improves as the size of K increases. However, our objective was not to find the ideal value of K but to identify a value that minimizes computational overhead and can increase the potential of the model. This was the rationale for fixing K at 5 across all experiments. We genuinely appreciate your precise observation and will ensure this point is addressed in our manuscript to aid our readers’ understanding. Thank you.
>
> | Model |Num K| R@1 | R@5 | MeanR |
> |-------|-------|--------|-------|-------|
> | X-CLIP | Baseline | 47.2 | 73.5 | 13.8 |
> | |K = 3| 48.1 | 73.2 | 13.5 |
> | |K = 5 | 48.4 | 73.7 | 13.2 |
> ||K = 10 | 48.4 | 73.3 | 13.3 |
> |CLIP4Clip| Baseline| 43.0 | 71.9 | 15.6|
> ||K = 3| 44.3 | 72.2 | 14.6 |
> ||K = 5|44.5 | 72.2 | 14.6 |
> ||K = 10| 45.0 | 72.3 | 15.0 |

---

> > ### Author Response · Authors · 2023-11-13
> > **Response to Reviewer Cekr (2/2)**
> >
> > >What if in some certain samples, there does not exist a scene transition or contains more than one transition? Can the proposed STMA be applied to all possible conditions?
> >
> > As detailed in Appendix A.6, our STMA has demonstrated the capability to adapt seamlessly, even in instances where scene transitions are absent or when multiple transitions are present. In cases when there isn’t any scene transition, the query and key within the Co-Attention mechanism would contain similar frames, leading to a learning pattern akin to that of a vanilla attention layer. Moreover, even with numerous scene transitions, the Co-Attention Layer allows the segmented $A$ scenes to attend to the corresponding $B$ scenes effectively. As illustrated in Figure 4, $X$ involves the attention of the "entire video" through a temporal encoder, which prioritizes important frames before concatenation. Subsequently, the alignment attention, which operates similarly to general attention mechanisms, refines the focus as depicted in Figure 9, ensuring the model's attention is directed towards the most relevant scenes. We have endeavored to ensure our model's robustness across various scenarios, and the empirical results support its effectiveness.
> >
> > ---
> >
> > Thank you for your time and consideration.
> >
> > Best regards,

---

> ### Author Response · Authors · 2023-11-21
>
> Dear reviewer Cekr,
>
> We are grateful for your constructive feedback.
>
> We would appreciate informing us whether your concerns have been adequately addressed.
>
> Your insights are highly valued, and we would like to let you know that we would happily answer any further questions if any.
>
> Best regards, The Authors

---

### Official Review · Reviewer_tWWe · 2023-11-02

**Soundness:** 2 fair
**Presentation:** 3 good
**Contribution:** 2 fair
**Rating:** 5
**Confidence:** 3

**Summary:**

This work tackles the problem of text-to-video retrieval and mainly focuses on the methods based on the pretrained CLIP. To mitigate the overfitting onto a target video dataset, this work proposes a two-stage training method where the first stage optimizes the image-to-video weight transfer, and the second stage introduces an adaptor to further improve the video understanding. The proposed method PATHS is a plug-and-play module and can be added to other existing methods such as CLIP4Clip, X-CLIP, DiCoSA. When tested on MSVD, LSMDC and MSRVTT retrieval benchmarks, PATHS improves over different baseline methods.

**Strengths:**

* The proposed PATHS method is a plug-and-play model that can be added to other existing methods and consistently improves the retrieval performances.
* The code is available which helps the reproducibility of the method.

**Weaknesses:**

* As the method requires more frequent evaluation than the standard per-epoch evaluation, the method still introduces computation overhead and leave the frequency as a hyperparameter which will potentially vary for different datasets.
* The contribution of the STMA adaptor seems marginal. For example in Table 1, the gain from STMA is only 0.2 point.
* In the ablation section, all BP, SP, USP methods exhibit very similar results. It is hard to tell if one quantifying strategy is better than others, raising a question whether the quantifying strategy is a main component of the proposed method.
* Typo "raking" --> "ranking" in section 5.4

**Questions:**

* While the proposed method focuses on the retrieval task, would PATHS method also applicable to other video-text tasks such as video captioning or video QA?
* Please see the weaknesses section for other questions.

---

> ### Author Response · Authors · 2023-11-13
> **Response to Reviewer tWWe**
>
> First of all, please accept our apologies for submitting the appendix through supplementary material only, which might have been missed as it was not appended to the main draft.
>
> We rectified this error by updating the submission to include the appendix within the manuscript. We are deeply sorry for any inconvenience this has caused and would be grateful if you could take a look at the appendix with reconsideration.
>
> We sincerely thank you for the constructive comments. We address the concerns below.
>
> ---
>
> >As the method requires more frequent evaluation than the standard per-epoch evaluation, the method still introduces computation overhead and leave the frequency as a hyperparameter which will potentially vary for different datasets.
>
> Your concern regarding the computational overhead due to more frequent evaluations is indeed valid. However, our perspective is to address the limitation of conventional epoch-wise evaluation, which may not fully use the potential of “existing” models. Our goal is to achieve higher performance while keeping the computational overhead minimized. We believe that while the ideal scenario would be to obtain optimal results with equal or reduced computational costs compared to epoch-wise evaluation, our method demonstrates enhanced efficiency against the current alternative ($N-step$ evaluation). We aim to emphasize that our approach is designed to strike a balance between uncovering the potential of models and maintaining computational feasibility.
>
> While the frequency ($K$) is a hyperparameter, we have fixed it to $K=5$ across all the datasets in our experiments without any further tuning. We claim that the reported results are not achieved through extra parameter tuning.
>
> ---
>
> >The contribution of the STMA adaptor seems marginal. For example in Table 1, the gain from STMA is only $0.2$ point.
>
> We appreciate the reviewer’s observation regarding the perceived marginal contribution of the STMA adapter, as reflected in Table 1. However, it is basically designed as an adapter for transfer learning. If only STMA is applied in stage 1, the STMA module may not seem relatively important. This is because the complexity of the model increases by adding an adapter while the model is not optimized. However, it is different in stage 2. For $w/o STMA$ in Table 1 at stage 2, $R@1$ performs well, but you can see that $R@5$, $R@10$, and $MeanR$ are overfitted compared to the baseline. Our parameter-wise two-stage method pivots based on the $R@1$ performance on the validation set during the second stage, which could indeed lead to overfitting on $R@5$, $R@10$, and $MeanR$. To counteract this, we integrate the “scene” information into the model learning process, which not only mitigates the risk of overfitting but also contributes to an incremental increase in $R@1$.
>
> ---
>
> >In the ablation section, all BP, SP, USP methods exhibit very similar results. It is hard to tell if one quantifying strategy is better than others, raising a question whether the quantifying strategy is a main component of the proposed method.
>
> Thank you for your insightful comments regarding the ablation study and the performance of $BP$, $SP$, and $USP$ methods. It's worth noting that these methods are employed to determine the parameter ranking within our framework. We do not explicitly confirm one strategy over others, as we intend to provide a flexible approach that can accommodate various quantifying methods. We believe many viable methods could be utilized effectively in this context. Our presentation of these methods aims to highlight the empirical observation that monitoring the extent of parameter rank fluctuations can indicate performance improvements at most rank fluctuation points. By showcasing these strategies, we suggest a direction for future research to quantify parameter changes to understand and optimize model performance. This part of our work offers a groundwork for further strategies to develop and refine.
>
> ---
>
> >Typo "raking" --> "ranking" in section 5.4
>
> Thank you for pointing out the typographical error. Your attention to detail is greatly appreciated, and we will ensure the correction is made in the revised manuscript.
>
> ---
>
> >While the proposed metod focuses on the retrieval task, would PATHS method also applicable to other video-text tasks such as video captioning or video QA?
>
> We thank the reviewer for brining this up; PATHS can be applied to other tasks, where we have tested for video QA in Section A.1 in the appendix. We regret that the appendix wasn't appended at the end of main draft, but uploaded separately through supplementary material.  We apologize for any inconvenience.
>
> ---
>
> Thank you for your time and consideration.
>
> Best regards,

---

> ### Author Response · Authors · 2023-11-21
>
> Dear reviewer tWWe,
>
> We are grateful for your constructive feedback.
>
> We would appreciate informing us whether your concerns have been adequately addressed.
>
> Your insights are highly valued, and we would like to let you know that we would happily answer any further questions if any.
>
>
> Best regards,
> The Authors

---

### Meta-Review · Area_Chair_hQwx · 2023-12-11

**Metareview:**

Among the five reviewers who submitted the comments, three tend to reject this work. Their main concerns are: the key concept proposed in this paper is tricky, and moreover, the strategy used in this work can potentially compromise robustness in real-world applications. Also, they pointed out that as the paper is not well written and organized, there are lots of confusions in this work. Besides, the performance gain brought by the designed techniques seems to be marginal. Considering these weaknesses, the AC believes this paper is not ready for publication at this stage, and encourages the authors to carefully address the reviewers' suggestions for future submission.

**Justification For Why Not Higher Score:**

The paper is not well written, which leads to confusions in this work. The method design is also tricky, limiting the contributions of this work.

**Justification For Why Not Lower Score:**

N/A

---

### Decision · Program_Chairs · 2024-01-16

Reject